# The Suitability of RNA from Positive SARS-CoV-2 Rapid Antigen Tests for Whole Virus Genome Sequencing and Variant Identification to Maintain Genomic Surveillance

**DOI:** 10.3390/diagnostics13243618

**Published:** 2023-12-07

**Authors:** Annamaria Cucina, Flavia Contino, Giuseppina Brunacci, Valentina Orlando, Mario La Rocca, Sergio Indelicato, Francesca Di Gaudio

**Affiliations:** 1Azienda Ospedaliera Ospedali Riuniti Villa Sofia-Cervello, Chromatography and Mass Spectrometry Section, Quality Control and Chemical Risk (CQRC), Via del Vespro, 133, 90127 Palermo, Italy; annamariacucina1@gmail.com (A.C.); f.contino@villasofia.it (F.C.); giuseppina.brunacci@villasofia.it (G.B.); v.orlando@villasofia.it (V.O.); 2Regional Health Department, Department of Strategic Planning, Piazza Ottavio Ziino, 24, 90145 Palermo, Italy; mario63larocca@virgilio.it; 3PROMISE-Promotion of Health, Maternal-Childhood, Internal and Specialized Medicine of Excellence “G. D’Alessandro”, University of Palermo, Piazza delle Cliniche, 2, 90127 Palermo, Italy

**Keywords:** COVID-19, SARS-CoV-2, rapid antigen tests, genomic surveillance, Omicron variant, viral sequencing

## Abstract

The COVID-19 pandemic has transformed laboratory management, with a surge in demand for diagnostic tests prompting the adoption of new diagnostic assays and the spread of variant surveillance tools. Rapid antigen tests (RATs) were initially used only for screening and later as suitable infection assessment tools. This study explores the feasibility of sequencing the SARS-CoV-2 genome from the residue of the nasopharyngeal swab extraction buffers of rapid antigen tests (RATs) to identify different COVID-19 lineages and sub-lineages. Methods: Viral RNA was extracted from the residue of the nasopharyngeal swab extraction buffers of RATs and, after a confirmation of positivity through a reaction of RT-PCR, viral genome sequencing was performed. Results: Overall, the quality of the sequences obtained from the RNA extracted from the residue of the nasopharyngeal swab extraction buffers of RATs was adequate and allowed us to identify the SARS-CoV-2 variants’ circulation and distribution in a period when the use of molecular swabs had been drastically reduced. Conclusions: This study demonstrates the potential for genomic surveillance by sequencing SARS-CoV-2 from the residue of the nasopharyngeal swab extraction buffers of RATs, highlighting alternative possibilities for tracking variants.

## 1. Introduction

The COVID-19 pandemic has significantly impacted laboratory management over the past three years. The rush in demand for diagnostic tests led to the rapid adoption of new diagnostic assays, and in order to track emerging variants and better understand viral transmission, it was necessary to also implement viral genomic surveillance through sequencing the SARS-CoV-2 genome. The first SARS-CoV-2 genome sequences were published in January 2020 [1]. Until real-time reverse transcription polymerase chain reaction (RT-PCR) became the most widely used approach early in the pandemic, SARS-CoV-2 genome sequencing had been applied to positive diagnostic samples from RT-PCR. Currently, reverse transcription polymerase chain reaction (RT-PCR)—a nucleic acid amplification test (NAAT)—is the gold-standard test for the diagnosis of SARS-CoV-2 according to the World Health Organization (WHO) and Centers for Disease Control and Prevention (CDC) [2]. RT-PCR offers high sensitivity and specificity due to the use of specific primers. However, it may not be suitable for emergency screening or surveillance applications due to its cost, time-consuming nature, and the need for expensive equipment and well-trained personnel. Additionally, RT-PCR cannot differentiate between active infection and residual nucleic acids from inactivated viruses, leading to false-positive results. Moreover, it faces challenges in detecting highly mutated variants, potentially resulting in false-negative outcomes. Despite these limitations, RT-PCR remains the reference method for diagnosing COVID-19 [3,4].

Rapid antigen tests (RATs) have become a valid option for detecting SARS-CoV-2. These tests offer significant advantages, such as speed, ease of use, cost-effectiveness, and suitability for both laboratory personnel and the public, which has led to their extensive diffusion. RATs are effective in detecting high viral concentrations in samples, making them especially useful in the early days of symptom onset. They are also useful for COVID-19 screening in large populations, as they can identify specific viral proteins (antigens) in 15–40 min via anti-SARS-CoV-2 monoclonal antibodies [5]. Various types of chromatography-based SARS-CoV-2 antigen tests are available, including enzyme immunoassays (EIAs), fluorescent immunoassays (FIAs), and chemiluminescence enzyme immunoassays (CLEIAs), which are laboratory-based methods, as well as rapid lateral-flow immunoassays (LFIAs) [6,7]. While RATs exhibit excellent specificity, they may not match molecular assays in terms of sensitivity. Nevertheless, according to the WHO, RATs provide a less resource-intensive and rapid means of SARS-CoV-2 detection, with a recommended specificity of at least 97% and sensitivity of at least 80% compared to a nucleic acid amplification test (NAAT) reference assay for healthcare usage. The FDA also approves RAT kits for COVID-19 diagnosis with a minimum sensitivity of 80%.

SARS-CoV-2 genome sequencing has been extensively applied to positive diagnostic samples obtained from NAATs. The gold standard and most commonly used NAAT is RT-PCR. Since both viral sequencing and RT-PCR involve amplification of viral genetic material, there is a significant overlap in collection methods, reagents, and downstream protocols, making it a valuable approach for genomic surveillance. This approach was particularly effective in the early stages of the pandemic. However, the testing landscape has since evolved, with a shift towards RATs, especially antigen-based lateral flow tests (LFTs). While antigen-based LFTs may have lower sensitivity compared to NAATs, especially in cases of low viral load or asymptomatic infection, they can achieve high sensitivity (99.2%) and specificity (100.0%) when used within 5–7 days of symptom onset among symptomatic individuals [8]. LFTs are efficient when the viral loads correspond to RT-qPCR Ct values equal or inferior to 33 cycles [9,10,11].

After the first wave of the COVID-19 pandemic, numerous variants of concern (VOCs), such as Alpha (B.1.1.7), Beta (B.1.351), Gamma (P.1), and Delta (B.1.617.2), were identified in South Africa, Brazil, and India. In November 2021, the Omicron variant was identified in South Africa [12]. This variant was rapidly classified as a VOC due to the presence of several mutations that could impact the transmissibility and severity of the disease, and immune escape [13]. The Omicron variant was reported to cause a significantly lower proportion of moderate or severe diseases leading to the deaths of the infected individuals compared to the Delta variant [12,14]. Nevertheless, the Omicron variant has warranted an urgent global public health alert due to its high contagiousness and vaccine-evading properties [13]. Furthermore, the Omicron variant, remarkably, has evolved into multiple sub-variants including BA.1, BA.2., BA.3, BA.4, and BA.5 [13,14].

With the emergence of the Omicron variant, both RT-PCR and RATs have been extensively evaluated to ensure accurate COVID-19 diagnosis. Omicron’s numerous mutations have led to changes in protein conformation, affecting the binding affinity of antibodies to lateral flow antigen tests. This has raised concerns about potential reductions in sensitivity compared to other variants. The FDA has indicated that RATs may have reduced sensitivity in detecting the Omicron variant, and studies assessing the reliability of RATs for Omicron detection have produced conflicting results, possibly due to the presence of undiagnosed Omicron-infected patients [15,16,17,18,19,20,21].

Previous works tested the possibility of recovering and sequencing viral RNA extracted from swabs or cassettes previously used for RATs by using a second buffer of extraction [22,23]. The same possibility was investigated by extracting the viral RNA directly from an excised part of the RAT strip [24]. However, these studies, which required additional steps for the recovery and extraction of the RNA material, have not been extended to a large population. In this context, considering that the loss of sensitivity to Omicron and other variants of RATs is linked to the detection technique but not to the sampling and extraction methods, and given that these are widely used, the present work aims to examine the possibility of sequencing the virus RNA extracted directly from the residue of the nasopharyngeal swab extraction buffers of positive RATs.

## 2. Materials and Methods

### 2.1. Collection of Samples and Participants’ Recruitment

The nasopharyngeal swab extraction buffers from the RAT devices that tested positive for SARS-CoV-2 during the Mediterranean Fiera Hub screening in Palermo, Sicily, where antigen testing was performed as part of routine clinical and public health practice, were collected from June 2022 to December 2022. The RATs used were the Fluorecare^®^ SARS-CoV-2 Spike Protein Test Kit (Microprofit Biotech, Shenzhen, China), which is classed as a medical device (CE IVD certificate). It is an immunochromatography assay used to determine the presence of the SARS-CoV-2 Spike protein which, if present, forms an immune complex evidenced by specific control (C) and test (T) bands.

It presents a declared sensitivity of 88%, referring to Wuhan variant proteins.

A total of 1282 residues of the nasopharyngeal swab extraction buffers, which remained after the antigen testing, were collected, and nucleic acid extraction was performed as later described, with processing started on the day of receipt of the sample at the laboratory. After collection, the material was stored at 4 °C.

### 2.2. SARS-CoV-2 RNA Extraction and RT-qPCR

For all samples, the entire residual volume of swab extraction buffers was diluted with 200 µL of phosphate-buffered saline and was used as the starting material for extraction in the MagMAX Viral/Pathogen Nucleic Acid Isolation Kit, performed on a KingFisher Apex instrument (Thermo Fisher Scientific, Norristown, PA, USA) according to the manufacturer’s instructions.

After extraction, eluates were tested and amplified on the QuantStudio™ 5 Real-Time PCR System (Applied Biosystems, Waltham, MA, USA) using the Thermo Fisher^®^ TaqPath™ COVID-19 CE-IVD RT-PCR Kit. The TaqPath assay targets three sequences in the virus: ORF1a, ORF1b, and N genes. The internal control for nucleic acid extraction was the RNaseP gene.

### 2.3. SARS-CoV-2 Viral Genome Sequencing

Libraries were prepared using the Illumina COVIDSeq protocol (Illumina Inc., San Diego, CA, USA). Briefly, first-strand cDNA was synthesized using reverse transcriptase and random hexamer primer. The SARS-CoV-2 genome was amplified using two sets of primers in two multiplex PCR protocols. Libraries were constructed by tagmentation and adapter ligation using IDT (Integrated DNA Technologies, San Diego, CA, USA) for Illumina Nextera UD Index Set A. Individual libraries were quantified using a Qubit 4.0 fluorometer (Invitrogen, Inc., San Diego, CA, USA) and pooled in an equimolar concentration. The quality of libraries was also checked on an Agilent TapeStation 4200 (Agilent, Santa Clara, CA, USA cat # G2991BA). Normalized library pools were sequenced using a High Output Flow Cell (600 cycles) Kit v3 on an Illumina MiSeq^®^ instrument. Illumina BaseSpace (https://basespace.illumina.com) bioinformatics was used for data QC, FASTQ generation, genome assembly, and SARS-CoV-2 variant detection. Briefly, raw FASTQ files were trimmed and quality-checked (Q > 30) using the BaseSpace FASTQ-QC application. QC-passed FASTQ files were aligned to the SARS-CoV-2 reference genome (NCBI reference sequence NC_045512.2). BaseSpace DRAGEN COVID Lineage was used to determine the SARS-CoV-2 variant and to generate a single consensus FASTA file.

### 2.4. Quality Evaluation with I-Co-Gen

The Istituto Superiore di Sanità (ISS) coordinates a network of laboratories across all Italian regions to systematically collect genetic sequences of SARS-CoV-2. These sequences are consolidated within a centralized national platform named ITALIAN-COVID-19-GENOMIC (I-Co-Gen). Accredited laboratories within each region have privileged access to this platform, facilitating the comprehensive collection, analysis, and dissemination of genomic characterization data related to SARS-CoV-2 at both regional and national scales. The database architecture is compartmentalized into regional projects, establishing discrete data repositories for each specific region. I-Co-Gen accommodates submissions of genomic sequencing data, including sequences related to the Spike protein-encoding gene. Employing automated algorithms, the platform conducts intricate analyses, providing results to the user while simultaneously archiving them in a national repository. Notably, this repository incorporates an automated alert system, triggered by the identification of variants of concern pertinent to public health or the emergence of novel variants. The user initiates the process by generating a sample and uploading next-generation sequencing reads. Subsequently, an automated analysis pipeline is activated, executing a sequence of typing and clustering operations. These operations terminate with the identification of genomic variants and a qualitative evaluation of the sample sequence. Moreover, I-Co-Gen is linked to the international sharing platform GISAID (Global Initiative on Sharing Avian Influenza Data), which facilitates rapid and unrestricted access to data concerning epidemic and pandemic viruses.

## 3. Results

### 3.1. RAT Sequencing and Quality Control of Sequencing with i-Co-Gen

Considering all the 1367 positive samples, regardless of their Ct, 1282 passed the sequencing (94%). Of the 1282 passed samples, 1119 (87%) were of high quality according to the quality evaluation of the sample sequences performed by I-Co-Gen (Figure 1). Detailed information about Ct value, identified lineages, and quality evaluation for each month are reported in Appendix A.

The comprehensive analysis of the sequencing data obtained from the residue of the nasopharyngeal swab extraction buffers of the rapid antigen tests revealed significant insights into the dynamics of Omicron variant sub-lineages over the course of several months. Specifically, the results depicted in Figure 2 indicated that in the month of June, Omicron 2 held a predominant position, constituting 54% of the total sequences obtained through RNA amplification of the residue of the nasopharyngeal swab extraction buffers. However, a remarkable shift in the prevalent sub-lineage occurred in July, with Omicron 5 emerging as the dominant variant, a position it maintained until December.

Interestingly, Omicron 5 exhibited a peak in its prevalence in the month of August, where it constituted 92% of the total sequences analyzed. Its presence gradually diminished, although it remained substantial, accounting for approximately 73–75% of the sequences in the months of November and December.

When considering the overall period from June to December 2022, the sequencing analysis unequivocally demonstrated the supremacy of the Omicron BA.5 sub-variants. This particular sub-lineage was remarkably prevalent, encompassing a significant 77% of the sequences obtained through RNA amplification of the residue of the nasopharyngeal swab extraction buffers.

Our analysis of SARS-CoV-2 variants over the specified months revealed noteworthy trends. Detailing those for each month, in June, Omicron 5 dominated at 37.07%, with a significant presence of Omicron 2 (54.31%) and Omicron 4 (6.90%). In July, the presence of Omicron 5 increased (89.31%), and the presence of Omicron 2 (4.40%) and Omicron 4 (4.40%) reduced. August and September displayed a consistently high prevalence of Omicron 5 (91.94% and 88.56%, respectively), with diminishing Omicron 2 and Omicron 4. October showed a diverse variant profile of distribution, with a notable emergence of recombinant variants (5.73%) at the expense of Omicron 4 (2.08%), while the substantial Omicron 5 (87.50%) and Omicron 2 (2.60%) presence remained unchanged. In November, Omicron 5 remained prevalent at 72.90%, while Omicron 2 reduced to 7.74%. Recombinant variants increased significantly to 18.71%, indicating a dynamic shift. December displayed similar patterns, with Omicron 5 at 74.74%, Omicron 2 at 11.58%, and recombinant variants at 13.68%. These findings highlight the evolving landscape of SARS-CoV-2 variants, emphasizing the dominance of Omicron 5 and the emergence of recombinant forms during the studied period.

The assessment of sequence quality, a critical aspect of our study, delved into the characteristics of the sequencing obtained from the residue of the nasopharyngeal swab extraction buffers of RATs by employing stringent Illumina quality criteria. Notably, the second level of quality (I-Co-Gen) evaluation exhibited its utmost reliability when the Ct values were confined within a narrow range (Ct < 28). Even within such stringent parameters, 10% of sequences faced second-level quality control failure. To visualize the monthly evolution of our findings, Figure 3 provides a comprehensive overview, detailing the percentage of sequences that presented high quality as well as those that presented low quality, along with the total number of positive swabs validated by the Illumina sequencing.

### 3.2. Variant Composition Comparison with ISS Data

The comparison between our research findings and the data reported by the Italian Institute of Health (ISS) for the period spanning June to December 2022 showed evident similarities in the variants’ distribution. A noteworthy observation that emerged from our study was the remarkably close resemblance of the monthly variant distributions observed through viral RNA obtained from the residue of RAT extraction buffer sequencing to those charted by the ISS for Italy (Figure 4). However, as with any dynamic system, subtle deviations occasionally emerged, providing valuable insights into the nuanced behavior of viral variants. In detail, in June, BA.2 was the dominant strain among our samples, accounting for 54.31% compared to the ISS’s 62.98%. However, we noted a substantial presence of BA.5 (37.07%), which exhibited differences from the ISS’s data (23.15%). July saw a notable rise in BA.5 prevalence in both datasets, reaching 89.31% in ours and 75.53% in the ISS’s data. BA.5 continued its dominance in August, comprising 91.94% of our samples and 90.79% according to the ISS. September demonstrated a similar trend, with BA.5 representing 88.56% in our dataset and 94.41% in the ISS’s. October showed a consistent prevalence of BA.5 at 87.50% in our samples and 93.01% according to the ISS, with minor variations in other sub-lineages. November revealed intriguing disparities: while BA.5 was predominant in both datasets, the data showed a difference of around 20% in the diffusion of this variant (72.90% in ours and 91.53% in the ISS’s) due to a substantial presence of recombinant variants in our samples (18.71%) compared to the ISS ones (2.39%). In December, BA.5 remained prominent, accounting for 74.74% in our data and 90.60% in the ISS’s. Additionally, we observed a significant but decreasing proportion of recombinant variants (13.68%), diverging notably from the ISS’s 3.00%.

### 3.3. Sub-Lineage Characterization

One of the remarkable outcomes of our sequencing efforts was the identification of a diverse array of lineages and sub-lineages. By sequencing viral RNA obtained from extraction buffers, we identified multiple variants, with a total of 30 lineages, along with even more sub-lineages, most of which were notably observed within the Omicron 5 variant, as illustrated in Figure 5.

Among the identified lineages, BN.1 emerged as the predominant lineage for Omicron 2. This lineage, marked by specific genetic signatures, provided valuable information about the evolutionary trajectory of Omicron 2. Conversely, when it came to Omicron 5, the predominant lineage was identified as BQ.1 (13.69%).

Within Omicron 5, we observed further genetic diversification, including the emergence of distinct subtypes, such as BE.1 (5.98%), BF.1 (1.72%), BF.5 (1.52%), and BF.7 (5.98%). Additionally, our analysis revealed the presence of rare subtypes, such as BE.2 (0.10%), BE.3 (0.10%), and DB.2 (0.20%), highlighting the complexity of Omicron 5 sub-lineages.

In addition to the well-defined lineages, our sequencing also brought to light the presence of recombinant variants, a phenomenon indicative of the intricate genetic interplay occurring within SARS-CoV-2. Notably, three distinct recombinant variants were detected, each offering a unique perspective on viral evolution. Among these, XBB emerged as the most prevalent recombinant variant, constituting a substantial 80% of the recombinant identifications, followed by XBF, which constituted approximately 14% of the recombinant identifications. This variant, while less prevalent than XBB, remained a crucial component of the genomic mosaic observed within RATs. Furthermore, XAX, although identified to a lesser extent at around 5%, provided yet another layer of complexity to the genomic diversity of SARS-CoV-2.

## 4. Discussion

The COVID-19 pandemic has unquestionably led to a paradigm change in laboratory management, particularly concerning diagnostic testing, throughout the past three years. The persistent surge in demand for diagnostic assays, driven by the urgency to identify and contain the SARS-CoV-2 virus, has necessitated an unprecedented acceleration in the adoption of novel diagnostic methodologies [3].

Considering the rapid evolution of the viral genome, the surveillance of variant diffusion was and is necessary in the pandemic management. During the initial phase of the pandemic, according to the Italian rules, a molecular test had to follow a positive RAT for positivity confirmation. Thus, the surveillance of variant diffusion was performed on the sequencing of the extracted RNA from molecular tests, effectuated on patients with a previous positive RAT. This procedure required the performance of two nasopharyngeal swab tests for each subject, with the related number of consumables and reagents used. This approach involved high costs of materials and waste disposal in a period of reduced availability of commodities and resources. Furthermore, it decreased the patients’ compliance. Conversely, in the second phase of the pandemic, this management approach became unfeasible because of the characteristics of the emerging COVID-19 variants. In fact, the central turning point in the COVID-19 pandemic was the advent of the Omicron variant [6]. This variant, characterized by a multitude of mutations, concerned the public health governance and the scientific community due to its transmissibility. Even though this variant presented the potential to evade immune responses, its large diffusion made it necessary to apply simplified methodologies for screening and monitoring. In response to these challenges, RATs emerged as valuable tools in the fight against the pandemic. These tests, designed for rapid and straightforward application, offer an important alternative in conditions demanding immediate results, such as mass screenings and emergency situations. RATs operate on the principle of detecting specific viral proteins (antigens) within a matter of minutes, providing a viable alternative to the time-intensive nature of RT-PCR [5]. The ease of use and cost-effectiveness of RATs, and their ability to provide results to both laboratory professionals and the general public, have made them pivotal in the global efforts to control the virus spread. However, it is crucial to acknowledge that the sensitivity of RATs, while worthy, might not match the standards set by RT-PCR [3,4,5,6,7]. This disparity is especially relevant in cases involving individuals with low viral loads or those in the asymptomatic phase of the infection, where RATs might exhibit reduced efficacy [7]. In fact, the efficacy is influenced by various factors, including viral genetics and the specific design features of the test devices. Consequently, each RAT responds differently to viral mutations, owing to the inherent differences in their designs. Preliminary studies and reports presented a spectrum of outcomes, with some RATs demonstrating resilience in the face of Omicron’s mutations, while others exhibited reduced sensitivity [25,26,27,28,29,30]. Nonetheless, starting from June 2022, the Italian rules changed, and the RATs were deemed sufficient to attest to the SARS-CoV-19 positivity or negativity. For this reason, RATs became performed almost exclusively, and sequencing from the buffer extracts of the molecular tests became unfeasible. Thus, many laboratories tried to make the best of this situation, exploring the opportunity to perform viral genome sequencing from the residues of their nasopharyngeal swab extraction buffers of RATs.

In the period between June and December 2022, we assessed the applicability of RATs outside of their intended purpose, leveraging residual volumes from the residue of the nasopharyngeal swab extraction buffers, aiming to evaluate this alternative methodology for pandemic surveillance.

The data analysis unveiled a dynamic panorama, showing the prevalence of Omicron BA.5, a sub-variant that emerged as a dominant force in the evolving viral landscape [22].

Our sequencing data were plotted against the national data disseminated by the Italian Institute of Health (ISS). This comparison allowed us to discern subtle divergences and striking resemblances. A pivotal aspect of our study involved the characterization of various lineages and sub-lineages within the Omicron variant. The genetic diversity observed within Omicron included the emergence of recombinant variants. Noteworthy among these was the identification of specific lineages, such as BN.1 and BQ.1 within Omicron 2 and Omicron 5, respectively [31].

In addition to well-defined lineages, our study unraveled the presence of less common recombinant variants within extraction buffer samples from RATs. The identification of distinct recombinant variants included XBB, CBR, and XAX. These variants, while representing a minority within the broader landscape, provided evidence of the multifaceted nature of SARS-CoV-2 evolution.

Furthermore, our data showed the possibility of sequencing the variants with second-level high-quality results. More specifically, in June, around one-third of the tests were categorized as low quality, indicating potential inaccuracies. In July, the number of low-quality tests halved. However, considering that the extraction and conservation procedures have not been changed over time, the cause of these fluctuations cannot be determined. After a consistent improvement in the quality results in August, reflecting the effectiveness of testing procedures, from September to the end of the observed period, the standard quality was maintained at a largely unaltered level. These findings highlight the overall reliability of the methodology used, with a consistent majority of high-quality results across the studied period. This evaluation has to take into account that only RNA extracted from the residual buffers of positive RATs was studied, and thus these samples contained a high viral load. Nonetheless, their data ensured reliable results, meaning such samples are appropriate for virus diffusion monitoring.

Future studies may address certain limitations of the present work by exploring and comparing the false-negative rates of RATs through parallel sampling or performing the RT-PCR also on the residual of nasopharyngeal swab extraction buffers of negative RATs. Furthermore, the possibility of using various commercially available RATs should be investigated. Our current study demonstrates the successful sequencing of SARS-CoV-2 from the residue of the nasopharyngeal swab extraction buffers used in lateral flow tests (LFTs). In summary, using the residue of the extraction buffer from LFT swabs is feasible and provides comparable RT-qPCR Ct values, genome coverage, and sequencing success rates, proving the possibility of genomic surveillance via sequencing from the RATs residual volume of swab extraction buffers, and expanding the toolkit for tracking SARS-CoV-2 variants. However, even if our data showed the feasibility of this approach in the analyzed period, taking into account that only the samples from positive RATs were sequenced in this methodology, we stress that the ability of the RATs to detect the viral antigen in the presence of new variants should always be verified, considering the variants’ evolution.

The availability of over-the-counter RATs empowers the public to assess personal risk and make informed decisions, which may extend to other respiratory pathogens, such as influenza and respiratory syncytial virus (RSV), often diagnosed in outpatient settings. Thus, viral genomic surveillance must be adapted in the future to ensure comprehensive community-based sampling, highlighting the need for collaboration between institutions, clinicians, and review boards to retain clinical specimens for genome surveillance.

## 5. Conclusions

In the ever-evolving landscape of COVID-19, our study has provided insight into the challenges and opportunities presented by the Omicron variant. By leveraging the residual volumes of the nasopharyngeal swab extraction buffers, we have reported the feasibility of sequencing SARS-CoV-2 RNA, offering a unique perspective on viral dynamics. Our findings suggest the possibility, with appropriate RATs, to simplify SARS-CoV-2 surveillance protocols, as tested in the particular scenario of the Omicron variant and the evolution of sub-lineages.

Continuously refining sequencing techniques and methodologies is imperative as we move forward. The behavior of SARS-CoV-2 variants necessitates a multifaceted approach, integrating technological advancements and collaborative efforts across scientific communities. By unraveling the complexities of viral evolution, we pave the way for targeted and effective strategies in combating infectious diseases. In the current scenario, the present study has suggested the possibility of utilizing our resources in new and advantageous diagnostic protocols.

## Figures and Tables

**Figure 1 diagnostics-13-03618-f001:**
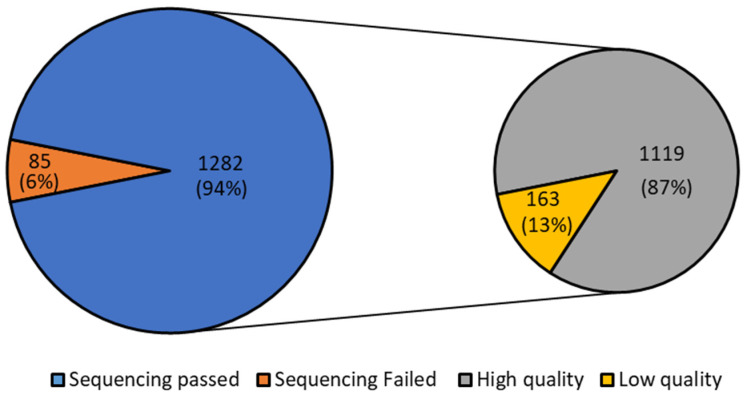
Pie chart of failure and passing of Illumina sequencing from June to December 2022 of the analyzed residue of the nasopharyngeal swab extraction buffers, and relative quality assessment.

**Figure 2 diagnostics-13-03618-f002:**
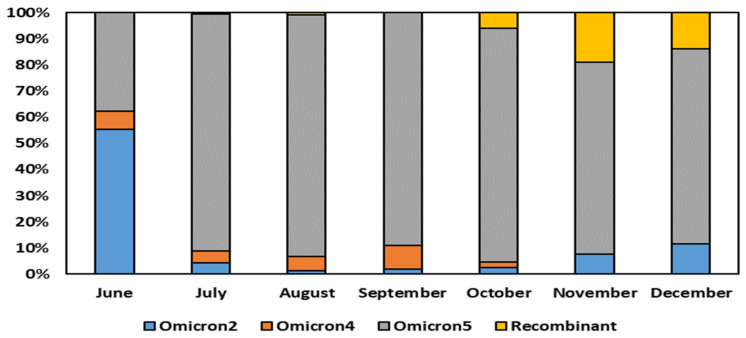
COVID-19 variant prevalence in the period June–December 2022.

**Figure 3 diagnostics-13-03618-f003:**
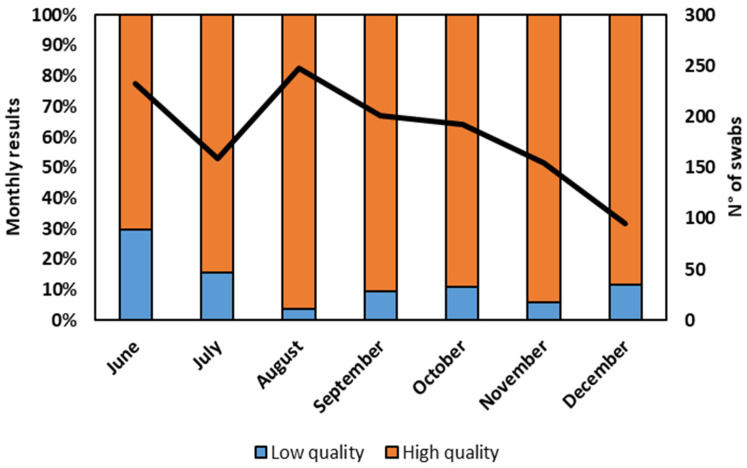
Failure and passing percentages of quality control sequencing from June to December 2022 and the number of analyzed residues of the nasopharyngeal swab extraction buffers for each month. The black line is the number of analyzed residues of the nasopharyngeal swab extraction buffers.

**Figure 4 diagnostics-13-03618-f004:**
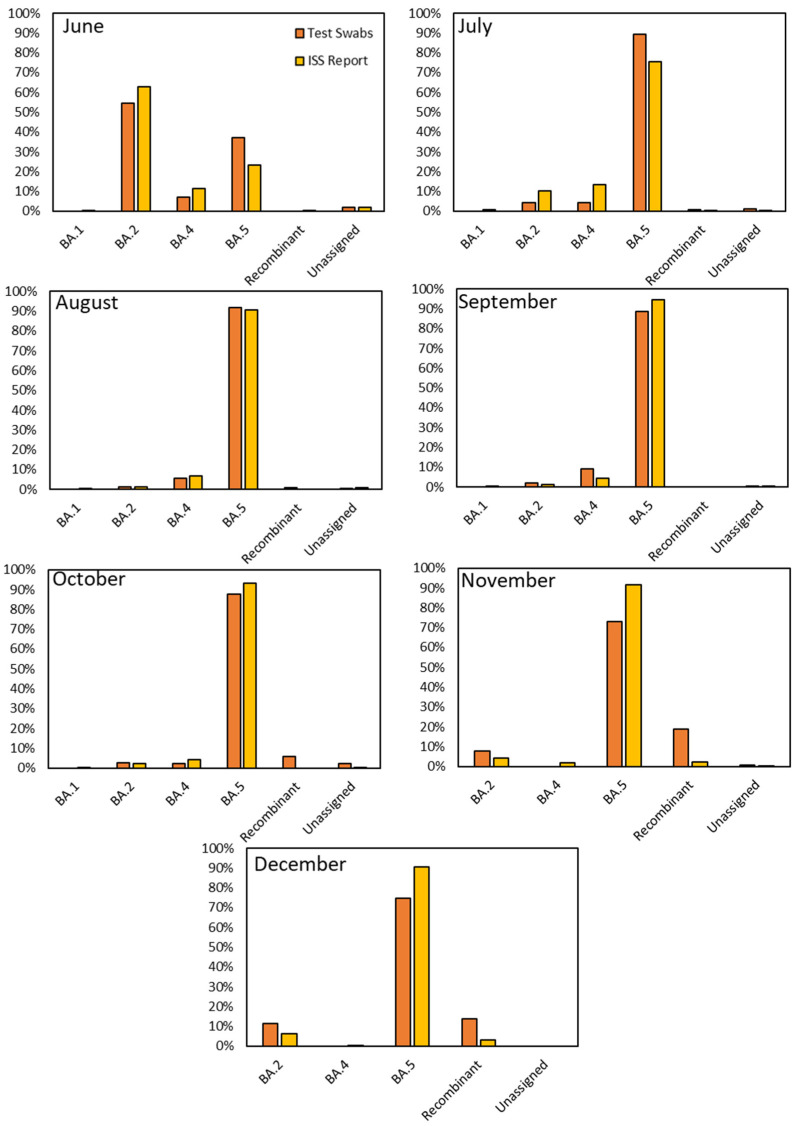
Comparison of variant prevalence between the RAT residues of the nasopharyngeal swab extraction buffers sequencing and national data from ISS in the period June–December 2022.

**Figure 5 diagnostics-13-03618-f005:**
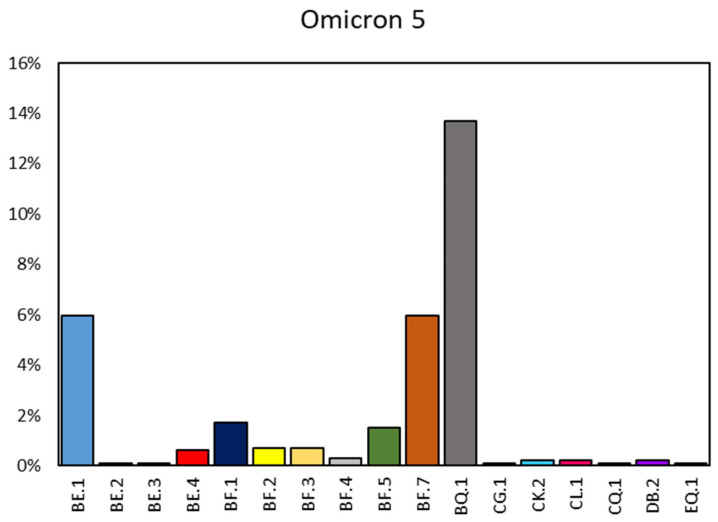
Omicron 5 lineages’ prevalence in the period June–December 2022.

## Data Availability

The data presented in this study are available on request from the corresponding author.

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
