# Peer review of "The Suitability of RNA from Positive SARS-CoV-2 Rapid Antigen Tests for Whole Virus Genome Sequencing and Variant Identification to Maintain Genomic Surveillance"

_diagnostics, 2023, doi:10.3390/diagnostics13243618_

Round 1

Reviewer 1 Report

Comments and Suggestions for Authors

Article to be rewritten.

The very title "Rapid Antigen Tests as Valuable Alternative for COVID-19 Omicron Variants Identification" raises strong objections, as it is misleading and not adequate to the presented study. RATs are not an alternative to determining virus variants.

What is the purpose of the study?

What has been presented is the use of material intended and used for RATs tests for an in-depth epidemiological analysis to the level of identifying virus variants. If such a procedure were applied continuously, we could talk about epidemiological monitoring of the infectious agent. The advantage is that we immediately select positive samples, just like with the RT-qPCR method. It is necessary to address the advantages and limitations of the material obtained in this way (costs, time, availability). In fact, at one point, pandemic monitoring switched to cheaper and faster antigen tests, so we obtained a broader and more reliable source of material with a much cheaper initial classification. On the other hand, RT-qPCR classification has a wider diagnostic window and sensitivity. Limitations in obtaining material classified on the basis of RATs test results can be minimized by the mass scale of tests performed and this should be emphasized, preferably with comparative data on the number of tests performed and their costs. It is also worth comparing the costs and time consumption of typing variants in routine diagnostics using methods other than sequencing. Should all laboratories switch to the presented procedure or only the reference one to detect newly emerging variants?

The authors themselves emphasize the diversity of RATs, which is why they must pay great attention not to generalize the conclusions to all RATs, but to limit them to those that have the characteristics of the test used, which has not been well characterized.

It would be interesting to compare the results of the antigen test and their confirmation by sequencing (determination of diagnostic accuracy parameters). This would provide characteristics of the test used in this study and this is what the authors should do. The source of the residual material itself does not justify exposing the method/test and devoting the entire discussion to it, and what was actually done does not stand out. For this reason, the structure of the article is greatly disturbed. The article needs to be thoroughly improved. In its current form, it was funny, but after correction, it must be checked by a diagnostician or someone who performs medical tests before re-posting, because the humor will not be appreciated.

At the end of the article, please complete: Author Contributions and Funding.

Detailed comment:

In line 69, RTDs used for the first time and further - line 364, are defined as "rapid diagnostic tests". In subsequent references: line 106, as "Rapid Detection tests". Please standardize.

Line 99-100. One may think that RDTs and RATs are two different techniques. Better to standardize.

Line 103 - What was the research material? How was it stored after collection?

Line 106 - Was the Fluorecare test set to Omicron or Wuhan variant proteins? Which variant did the given test sensitivity data (88%) refer to? Is the test a medical device (CE IVD certificate)? What validation has it undergone? This information is missing and critical.

Line 109-112. Previously, it was written that "RAT devices" were collected, and now that "nasopharyngeal swab extraction buffers" were collected. What's the truth ?

Line 163 – 87% of samples were sequenced in high quality ? In the chart from I-Co-Gen they are of low quality. What's the truth? The authors' wishful thinking or a mistake?

Line 168 – This is not a comprehensive analysis of RATs data. We do not receive sequencing data from RATs. Data from RATs are "positive" and "negative". The material from this buffer can be used for RT-qPCR and any other method of obtaining a result from this method. On this basis, the authors may well claim that this was a comprehensive analysis of RT-qPCR or IFA data as an alternative to identifying Omicron variants.

Line 170 – what is the time advantage to obtain results from this material source compared to another? The analysis took several months, so it is not a more favorable alternative to analyzes from RT-qPCR swabs.

Line 172 – There is no such thing as antigen test amplification.

Line 171 – 54% is not amazing. It constitutes approximately half of all variants.

Line 182 – “sequences obtained through rapid antigen test amplification”. Amplification of the test is possible as replication of the prototype on the production line. Unfortunately, sequencing RATs is impossible.

Line 199-203 – Fragment unintelligible. The authors refer to Ct as if they obtained it from antigen tests. What are these statements based on, where are the results and how were the measurements made?

Line 211 – what does improving testing standards mean and what does it result from? Please determine the cause of these fluctuations.

Figure 2 – “number of analyzed swabs” . What ultimately constituted the research material: tests, buffer or swabs?

Figure 4 – Unable to sequence tests. Fig's caption is irrational.

Line 227-228. The comparison is not obvious. Not everyone compares their data with ISS data. What is the purpose of this sentence?

Line 230 – authors are still sequencing RATs.

Line 253-255. “RATs demonstrated their ability to identify a staggering total of 30 lines […]”. Very interesting. If the RATs showed something, why weren't the results from these tests presented to support the claim? Line 340-342 .RATs do not predict variants. The result does not indicate a virus variant, so it cannot be claimed that it detects a specific variant, only that there may be such a variant in the detection profile. I would like to remind you to define and specify the detection components in the antigen test. What and what exactly does it detect?

Line 277 – The discussion did not refer the obtained results to those of other authors. This is just a rant about antigen tests. There is also no consideration of which antigen tests can be used for such surveillance and the benefits and limitations of this strategy compared to other strategies. The discussion did not address the aspect of the novelty of the study and its limitations - which must be taken into account.

Line 375-377 – “Our findings underscore the adaptability of RATs in capturing diverse SARS-CoV-2 lineages, with a particular focus on the Omicron variant and its sub-lineages.” What was the adaptation of the test for this application? What design changes were made?

Lina 371- Conclusion. What conclusions have been drawn from the results obtained from this work? What new has been created or used and what can be used by others? Most statements in conclusions are lofty, obvious generalities - they should be reformulated or limited to those that refer to the presented work.

Comments on the Quality of English Language

Moderate editing of English language required. It is mainly about the correct use of defined diagnostic concepts.

Author Response

Please find enclosed a revised version of our manuscript entitled “Rapid Antigen Tests as Valuable Alternative for COVID-19 Omicron Variants Identification.” (Manuscript ID: diagnostics-2665763).

As required (02-Nov-2023), together with the revised manuscript, where the modifications are highlighted in red, we now resubmit a point-to-point response to the reviewer’s criticisms (see the attachment).

We thank the Referees for their interest in our work and for helpful comments that greatly improved the manuscript. We have tried to do our best to respond to the points raised.

The Referees have brought up some good points and we appreciate the opportunity to clarify our research objectives and results. As indicated below, we have checked all the general and specific comments provided by the Referees and have made necessary changes accordingly to their indications.

Sincerely,
Francesca Di Gaudio

Reviewer 2 Report

Comments and Suggestions for Authors

The study's findings that the rapid antigen test can detect even sublineages of the SARS-CoV-2 variant may allay concerns about the use of RATs. Cross-reactivity of RATs with other respiratory tract samples may lead to false positives. The fact that it did not work with respiratory  samples that were not suspected of SARS-CoV-2 should be considered as a limitation of this study. However, especially in antigen tests, sample collection time is a factor that greatly affects sensitivity. How long after symptom onset did the authors obtain the samples? Is this situation reflected in the findings? More detailed information is needed about the antigen kit used in the material-method. Information on the rapid test method based on which method it is used and the manufacturer and its origin should be shared. When writing references, attention should be paid to the journal rules, especially when writing web references.                                                        

Comments on the Quality of English Language

It is only meed  a minor reision.

Author Response

(The authors gave the same response as above.)

Reviewer 3 Report

Comments and Suggestions for Authors

The manuscript authored by Cucina and colleagues presents findings that have culminated in the potential for genomic surveillance by sequencing SARS-CoV-2 from rapid antigen tests (RATs) swabs. In other words, the authors have investigated the possibility of successfully sequencing the virus RNA extracted from the swab of RATs and to investigate the capacity of the RATs to identify different lineages and sub-lineages.

From my perspective, this study holds relevance within the field, notwithstanding there is a need to make some modifications to improve the writing of the study. Thus, I have several points that I see interesting to add to this manuscript:

1. Abstract: I found the introduction and conclusion of the summary long. On the other hand, the methodology and results sections are short. As it stands, the methodology used and the results generated are not clear. I think it is important in these sections that authors seek to add value to their study.

2. Abstract (line 13): Remove the word "Background."

3. Introduction (line 31-33): I recommend the authors rewrite this paragraph to develop it further. Furthermore, no reference citation was added in this paragraph.

4. Results (line 162-164): It is important to make it clearer that the discrepancy between the positivity of the Fluorecare® SARS-CoV-2 Spike Protein Test Kit vs the samples that were positive for qPCR (QuantStudio). Of the 1367 screened for SARS-CoV-2 in the qPCR, 1282 were positive, correct?

5. Results (Figures): The font of the figures captions should maintain the same style as the main text sections of the manuscript.

 6. Results (line 224): This figure should be named Figure 3.

7. Discussion: For the information presented in the various paragraphs of this section, I believe it is extremely important for the authors to provide the source of this information by inserting references. Please review this for each sentence.

Author Response

(The authors gave the same response as above.)

Reviewer 4 Report

Comments and Suggestions for Authors

The article by Annamaria Cucina and co-authors is devoted to the topic of rapid diagnosis of coronavirus. This is a current topic, even though the incidence of diseases caused by this pathogen is on the decline. However, we must draw conclusions from the pandemic, including regarding diagnostic approaches.

The authors use rapid antigen tests in their work and also carried out genomic identification of virus variants. In particular, Omicron variants Overall they managed to show a clear picture.

The article contains the necessary illustrations. Their quality is beyond question.

The authors discuss the results obtained.

The overall impression of the work is very good.

Author Response

(The authors gave the same response as above.)

Round 2

Reviewer 1 Report

Comments and Suggestions for Authors

The authors have revised the manuscript insufficiently and in this form it is not suitable for publication.

There are still erroneous claims in the manuscript resulting from a misunderstanding of the main charge to the work presented. When the authors changed the title from: "Rapid antigen tests as valuable alternative for COVID-19 Omicron variants identification" to the correct "RNA from Positive SARS-CoV-2 Rapid Antigen Tests is Suitable for Whole Virus Genome Sequencing and Variants Identification for Maintaining Genomic Surveillance" one might have expected them to correct the article to the same extent, but they did not.

As in the original title, still in the manuscript the authors claim to identify viral variants with RATs tests.

Maintaining the test's specificity (no decrease in specificity) for successively emerging virus variants does not constitute their identification. For the average reader, the provisions formulated in this way clearly suggest the possibility of differentiating variants using RAT tests.

Examples:

Line 104-105 - "[...]capacity of the RATs to identify different lineages and sub-lineages."

The authors continue to state that the aim of the study was to identify lineages and sub-lineages of the virus by RATs testing. This is a serious factual error. RATs tests do not identify virus variants. The ability to detect a variant is not its identification.

Line 266-268 – “RATs demonstrated their ability to identify a staggering total of 30 lineages, with even more sub-lineages […]”

Line 357 - "In our research, the efficacy of RATs in detecting SARS-CoV-2, especially the Omicron variant and its sub-lineages [...]"

Line 359-362 - "The efficiency of rapid antigen tests (RATs) in detecting SARS-CoV-2, specifically the Omicron variant and its sub-lineages, is influenced by various factors, including viral genetics and the specific design features of the test devices."

RATs tests do not identify viral variants, so they also have no efficacy and efficiency in this regard.

Clear and correct wording throughout the article is a mandatory condition for acceptance of a manuscript for publication. If they are not corrected, the manuscript will be recommended for rejection.

After proofreading, the manuscript is more understandable, but I again suggest consulting a medical diagnostician.

Comments on the Quality of English Language

Minor editing of English language required.

Author Response

Thank you for your comment. We corrected the cited examples and cleared and corrected the wording throughout the article.

Furthermore, after consulting a medical diagnostician, we modified the Discussion and Conclusions sections to better highlight our goals and avoid misunderstandings.

We also improved our cited references.

Reviewer 3 Report

Comments and Suggestions for Authors

After revisions to the manuscript, the study underwent considerable improvement.

Author Response

Thank you for your comment.

We improved the cited references, including recent articles, in the Introduction section.

Round 3

Reviewer 1 Report

Comments and Suggestions for Authors

There are no longer any ambiguities in the latest version of the article. The article meets the criteria for publication in "Diagnostics".